# Pronounced Linewidth Narrowing of Vertical Metallic Split-Ring Resonators via Strong Coupling with Metal Surface

**DOI:** 10.3390/nano11092194

**Published:** 2021-08-26

**Authors:** Wei Du, Youcheng Zhu, Zhendong Yan, Xiulian Xu, Xiaoyong Xu, Jingguo Hu, Pinggen Cai, Chaojun Tang

**Affiliations:** 1College of Physics Science and Technology, Yangzhou University, Yangzhou 225002, China; wdu@yzu.edu.cn (W.D.); yczhu_yzu@163.com (Y.Z.); xuxl@yzu.edu.cn (X.X.); xxy@yzu.edu.cn (X.X.); jghu@yzu.edu.cn (J.H.); 2The National Laboratory of Solid State Microstructures, Nanjing University, Nanjing 210093, China; zdyan@njfu.edu.cn; 3College of Science, Nanjing Forestry University, Nanjing 210037, China; 4College of Science, Zhejiang University of Technology, Hangzhou 310023, China; caippgg@zjut.edu.cn

**Keywords:** magnetic plasmon resonances, split-ring resonators, ultranarrow-band hybrid mode, metamaterial

## Abstract

We theoretically study the plasmonic coupling between magnetic plasmon resonances (MPRs) and propagating surface plasmon polaritons (SPPs) in a three-dimensional (3D) metamaterial consisting of vertical Au split-ring resonators (VSRRs) array on Au substrate. By placing the VSRRs directly onto the Au substrate to remove the dielectric substrates effect, the interaction between MPRs of VSRRs and the SPP mode on the Au substrate can generate an ultranarrow-band hybrid mode with full width at half maximum (*FWHM*) of 2.2 nm and significantly enhanced magnetic fields, compared to that of VSRRs on dielectric substrates. Owing to the strong coupling, an anti-crossing effect similar to Rabi splitting in atomic physics is also obtained. Our proposed 3D metamaterial on a metal substrate shows high sensitivity (*S* = 830 nm/RIU) and figure of merit (*FOM* = 377), which could pave way for the label-free biomedical sensing.

## 1. Introduction

Plasmonic and metamaterial structures allow the coherent oscillations of free electrons, known as surface plasmons [1,2,3]. The resonant excitation of the localized surface plasmon resonances (LSPRs) and propagating surface plasmon polaritons (SPPs) concentrates light into subwavelength volumes and induces large electric field enhancements, known as ”hot spots”, which can be exploited in potential applications such as sensing [4,5], nonlinear optics [6,7], optical switching [8,9], photodetection [10,11], and solar energy absorbers [12,13] and related devices. Meanwhile, significant efforts have also been taken to explore nanostructures which are capable of providing localized magnetic enhancements [14,15,16,17,18].

Metamaterial composed of periodic arrays of sub-wavelength metallic split-ring resonators (SRRs) with the capability of enhancing magnetic field has been developed to give rise to novel electromagnetic properties and potential applications such as optical nonlinearity and magnetic biosensors [19,20,21,22,23,24]. Nevertheless, the magnetic resonance of the conventional planar SRRs usually has a relatively broad bandwidth and thus, a relatively weak enhancement of electromagnetic fields due to the fast radiation damping [25]. One effective method to enhance magnetic fields of the magnetic resonance and narrow the broad bandwidth is through coupling the MP resonance to other optical narrow-band resonance modes with high-quality factors, such as surface lattice resonances [26,27,28], Fabry-Perot cavity resonances [29,30], optical waveguide mode [31,32], or Tamm plasmons [33,34]. Chen et al. theoretically reported that the interactions between periodic metallic nanodisks and optical waveguide modes propagating in the adjacent dielectric waveguide lead to a narrow-band mixed mode with greatly enhanced magnetic fields, which can be tuned continuously by changing the array period [35]. 

Most of the planar SRRs are directly placed on dielectric substrates, which leads to a quite appreciable amount of electromagnetic energy spreading into the dielectric substrates and hampers the metamaterials sensing applications [20,21,36,37,38,39]. In recent years, vertical U-shaped SRRs on a dielectric substrate have been reported, in which the sensing medium can be fully spread into the free space of the hot spots of electromagnetic fields at the magnetic resonance [40,41]. This can be circumvented by removing the dielectric substrates and developing all-metal designs, allowing for the strong absorption of incident light through the excitation of magnetic hot spots [42,43,44,45].

In this work, we present an effective method to realize an ultranarrow-band hybrid plasmon mode with the full width at half maximum (*FWHM*) of 2.2 nm and greatly enhanced magnetic fields by plasmonic coupling between magnetic plasmon resonances (MPRs) and propagating surface plasmon polaritons (SPPs) in a three-dimensional (3D) metamaterial consisting of vertical Au split-ring resonators (VSRRs) array on Au substrate. By placing the VSRRs directly onto the Au substrate, to removing the effect of dielectric substrates, both the enhancement of the quality factor (*Q*) and magnetic field of the VSRRs hybrid plasmon mode are up to 5 and 2 up fold compared with those of VSRRs array on dielectric substrates. Moreover, an anti-crossing phenomenon similar to Rabi splitting in atomic physics, is also observed. Our proposed 3D metamaterials on the metal substrate have high sensing performance factors (*S* = 830 nm/RIU and *FOM* = 377), indicating its significant application potential in biomedical and sensing applications.

## 2. Materials and Methods

The designed 3D metamaterial is schematically depicted in Figure 1a. The array of Au VSRRs is directly on the Au substrate. Figure 1b shows the magnified front view of a unit cell of the Au VSRRs structure. The structural parameter of an individual VSRR: *l_x_* = *l_z_* = 90 nm and *w* = 20 nm. The periodicity *P* along *x* direction is set as 800 nm. The electric field *E*_in_, magnetic field *H*_in_ and wave vector *K*_in_ of the incident light are along the *x*, *y*, and *z* axes, respectively. Our proposed Au VSRRs structure is able to be achieved experimentally by the following process: Firstly, a thick Au film with a thickness of 100 nm is deposited on a glass substrate. Then, the Au VSRRs array is prepared by electron beam lithography with the double exposure process. We employ the commercial software package “EastFDTD, version 5.0” based on the finite-difference time-domain method to numerically simulate the plasmonic resonant behaviors of our designed Au VSRRs structure. In the *z* axis direction and the *x* axis direction, perfectly matched layers and periodic boundary conditions were applied, respectively. The permittivity of Au was calculated by using the Drude model [43] with the plasma frequency of *ω*_p_ = 1.37 × 10^16^ s^−1^ and the damping constant of *ω*_c_ = 4.08 × 10^13^ s^−1^.

## 3. Results and Discussion

Figure 2 shows the calculated normal-incidence reflection spectra of the designed array of Au VSRRs on Au substrate with the period *P* = 800 nm. The structural parameter of a single VSRR is the same as shown in Figure 1. For a normal incident transverse magnetic (TM) wave, a broad reflection dip (labeled as I) centered at 696 nm and an ultranarrow reflection dip (labeled as II) centered at 830 nm are observed, which are shown by the solid red line in Figure 2, respectively. The broad reflection dip arises from the excitation of magnetic resonances in an individual Au VSRR. More importantly, the ultranarrow reflection dip with its *FWHM* of 2.2 nm arises from the hybridization of propagating surface plasmon polaritons and magnetic resonances. For comparison, we also calculated the normal-incidence transmission spectra of Au VSRRs directly on silica substrate with the refractive index of 1.45 shown by the dotted blue line. The structural parameter of a single VSRR and the period are the same as those of Au VSRRs on Au substrate. There is a relatively narrow transmission dip (labeled as III) at 815 nm and a broad weak transmission dip on the left side of dip III at around 784 nm with the *FWHM* of 21 nm. Such an asymmetric Fano lineshape is due to the coupling between collective diffraction mode and the magnetic resonances of Au VSRRs on a silica substrate. Nevertheless, the Fano-like transmission window is weak and the bandwidth is much broader than the reflection dip II of Au VSRRs on Au substrate because of the reduced dielectric substrate effect.

Figure 3 shows the normalized electromagnetic field (*H*/*H*_in_ and *E*/*E*_in_) distributions on the *xoz* plane for the dip (I) and dip (II) of the Au VSRRs on Au substrate and for the dip (III) of the Au VSRRs on silica substrate, respectively. Obviously, both the magnetic field and electric field distributions of dip (II) are very similar to those of the dip (I) resonance, but the maximum magnetic field and the maximum electric field at the resonance of dip (II) are enhanced to be about 50 and 70 times of incident magnetic and electric fields, which are 2 and 2.33 times as strong as the corresponding values at the dip (III) resonance of the Au VSRRs on silica substrate.

In order to get a deeper insight into the nature of two hybrid modes of the Au VSRRs on Au substrate, the positions of reflection dips for different periods *P* increased from 550 nm to 1050 nm in steps of 50 nm are shown by the two branches of the open circles and red lines in Figure 4. We use a coupling model to investigate the coupling effects between MP resonance and the SPPs in the proposed metamaterials by the equation [2]: $E_{\pm }=(E_{\mathrm{MP}}+E_{\mathrm{SPPs}})/2\pm \sqrt{\Delta /2+{\left(E_{\mathrm{MP}}-E_{\mathrm{SPPs}}\right)}^{2}/4}$. Here, *E*_MP_ and *E*_SPPs_ are the excitation energies of the MP resonance and the SPPs, respectively. Δ represents the value of coupling strength. The black solid line denotes the MP resonance. For our proposed structure, the incident light wavelength to excite the SPPs under normal incidence can be calculated [2]: $\lambda_{\mathrm{SPPs}}^{n}=(P/n)\sqrt{\varepsilon_{m}/\left(\varepsilon_{m}+1\right)}$, where *n* is integer and *ε*_m_ is the relative permittivity of gold. The positions of reflection dips for different period *P* can be predicted accurately, as shown by the two branches of red lines in Figure 4. At the crossing of the SPPs and the MP resonance, the reflection dips exhibit an obvious anti-crossing similar to the Rabi splitting in atomic physics, indicating the strong coupling between the SPPs and the MP resonance. Such strong coupling in our proposed the Au VSRRs on Au substrate is able to generate an ultranarrow hybrid mode and a large electromagnetic field enhancement at the dip II resonance.

We next investigate the effects of the prong length *l_z_*, the base rod length *l_x_* and the width *w* shown in Figure 1b on two hybrid modes of the Au VSRRs on Au substrate. As shown in Figure 5a, under the condition of other fixed structural parameters (*l_x_* = 90 nm_,_
*w* = 20 nm and *P* = 800 nm), when the prong length *l_z_* increases from 80 nm to 120 nm in steps of 10 nm, both the reflection dips (I) and (II) show the obvious redshift. Meanwhile, the reflection intensity of the two reflection dips, (I) and (II) become weaker, and the bandwidth of the two reflection dips become broader as *l_z_* is increased. As shown in Figure 5b, when the base rod length *l_x_* increases from 80 nm to 120 nm in steps of 10 nm under the condition of other fixed structural parameters (*l_z_* = 90 nm_,_
*w* = 20 nm and *P* = 800 nm), the reflection dips (I) shows a blueshift while the reflection dips (II) shows a redshift. Both the reflection intensity of dips (I) and dips (II) are enhanced. Figure 5c exhibits that both the reflection dips (I) and (II) show the obvious blueshift when the width *w* increases from 10 nm to 30 nm in steps of 5 nm under the condition of other fixed structural parameters (*l_z_* = *l_x_*= 90 nm_,_ and *P* = 800 nm). Meanwhile, the reflection intensity of both the two reflection dips (I) and (II) become stronger as *w* is increased. 

Finally, we proceed to investigate the performance of the Au VSRRs on Au substrate for optical sensing. Figure 6a shows the calculated reflection spectra of our proposed 3D metamaterials immersed in different environmental media under the TM normal incidence while keeping geometrical parameters of the Au VSRRs on Au substrate is as the same as that of in Figure 2. For the refractive index of the environmental medium increased from 1.00 to 1.04 in intervals of 0.01, the two reflection dips (I) and (II) have obvious red-shifts. The dependence of the positions of the reflection dip II of the ultra-narrowband hybrid mode on the refractive index is shown in Figure 6b. By linearly fitting the data in Figure 6b, the refractive index sensitivity (*S*) [28] of the ultra-narrowband hybrid mode of the proposed Au VSRRs structure on Au substrate is obtained to be 830 nm/RIU. The figure of merit (*FOM*) is more meaningful to quantify the overall performance of a sensor, defined as the refractive index sensitivity divided by the resonance linewidth [28]. For the reflection dip II of the ultra-narrowband hybrid mode, its *FWHM* is 2.2 nm, and the *FOM* is achieved to be 377, which is enhanced up to 18 times as high as that of the reflection dip I of the MP resonance. The high *FOM* obtained in our proposed structure now reach nearly the highest values of the recently reported plasmonic sensors [46,47]. Such a good sensing capability of our proposed metamaterials is beneficial to highly sensitive detection of small changes of the refractive index of different environment media, which may have potential applications in biosensing.

## 4. Conclusions

In conclusion, we demonstrated a powerful method to realize an ultranarrow-band hybrid plasmon mode and greatly enhanced electromagnetic fields in a 3D metamaterial composed of VSRRs array on Au substrate. By placing the VSRRs directly onto the Au substrate to removing the effect of dielectric substrates, both the enhancement of the quality factor (*Q*) and magnetic field of the VSRRs hybrid plasmon mode are up to 5 and 2 up fold compared with these of VSRRs array on dielectric substrates through the strong plasmonic coupling between magnetic plasmon resonances and propagating surface plasmon polaritons. An anti-crossing phenomenon, similar to Rabi splitting in atomic physics, is also obtained. The ultra-narrowband hybrid mode of our proposed 3D metamaterials on metal substrate has an ultrahigh sensing performance factors (*S* = 830m/RIU and *FOM* = 377), indicating its significant application potential in biosensing applications. 

## Figures and Tables

**Figure 1 nanomaterials-11-02194-f001:**
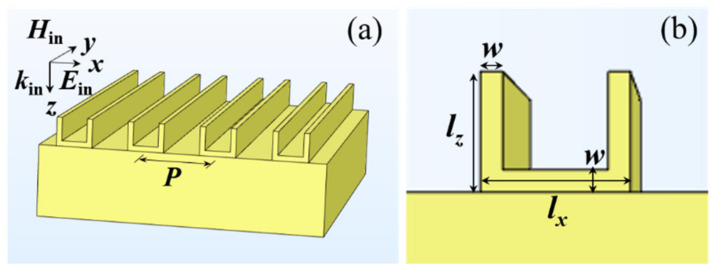
(**a**) Schematic view of the array of the Au VSRRs structure directly on the Au substrate. (**b**) The magnified front views of a unit cell of the Au VSRRs array.

**Figure 2 nanomaterials-11-02194-f002:**
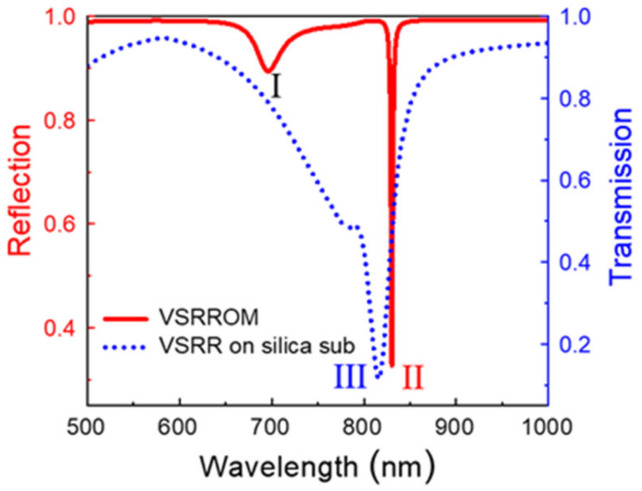
The calculated reflection of Au VSRRs on Au substrate and transmission of Au VSRRs on silica substrate. (I) denotes the MP resonance and (II) denotes the narrowband mixed mode of the Au VSRRs on Au substrate. (III) denotes the transmission dip of the SRR on a silica substrate.

**Figure 3 nanomaterials-11-02194-f003:**
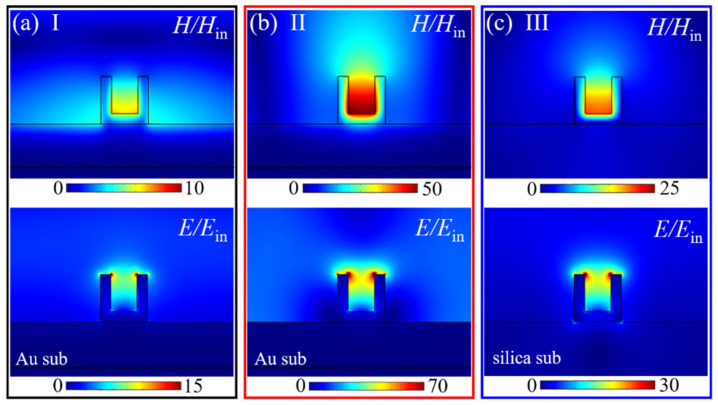
Normalized magnetic field (*H*/*H*_in_) and electric field (*E*/*E*_in_) distributions on the *xoz* plane for the dip I (**a**) and dip II (**b**) of the Au VSRRs on Au substrate and for the dip III (**c**) of the Au VSRRs on silica substrate.

**Figure 4 nanomaterials-11-02194-f004:**
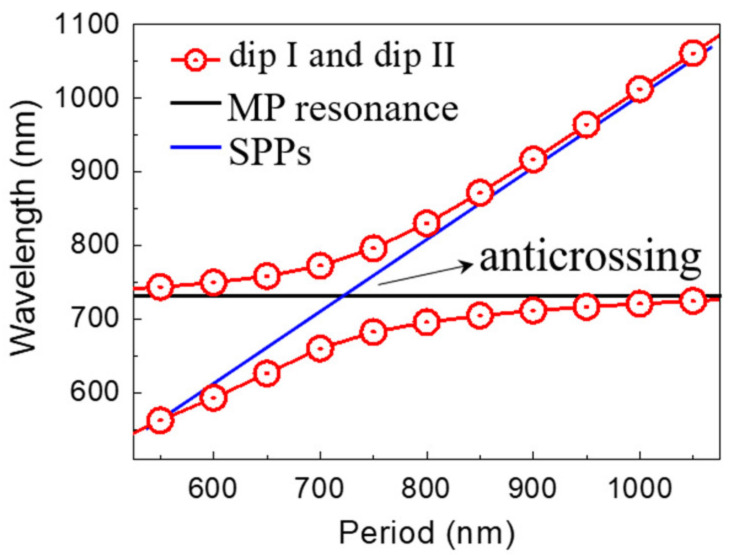
The dependence of the reflection dip positions (I) and (II) on the period *P*. The positions of the MP resonance (solid black line) and the SPPs (solid blue line) are also presented.

**Figure 5 nanomaterials-11-02194-f005:**
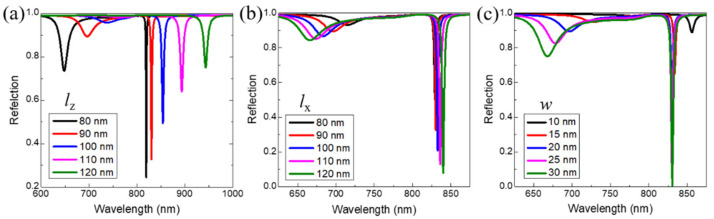
The calculated reflection spectra of the Au VSRRs on Au substrate (**a**) for different prong length *l_z_* (**b**) for different length of base rod *l_x_* at normal incidence and (**c**) for different width *w* of the Au split-ring resonators.

**Figure 6 nanomaterials-11-02194-f006:**
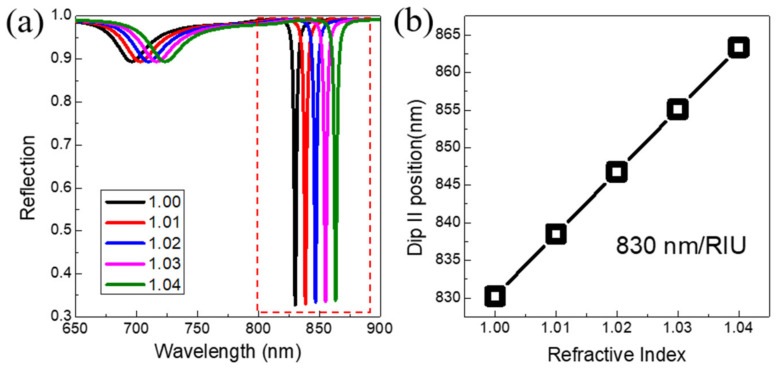
(**a**) The calculated normal-incidence reflection spectra of the Au VSRRs on Au substrate with TM polarization immersed in different environment media. (**b**) The resonance wavelength of the dip II extracted from (**a**) as a function of refractive index.

## Data Availability

The data presented in this study are available on request from the first or corresponding author.

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
