# Peer review of "Pronounced Linewidth Narrowing of Vertical Metallic Split-Ring Resonators via Strong Coupling with Metal Surface"

_nanomaterials, 2021, doi:10.3390/nano11092194_

Round 1
Reviewer 1 Report
The paper is interesting, well written and clear.
Since only numerical simulations are reported, a 3D analysis with finite dimensions along y-direction could be provided to validate the results.
The effects of varying the width w of the Au split-ring resonators can be considered.
Reviewer 2 Report
Reviewer comments for nanomaterials-1345138 by Y. Chen, et al, titled “Pronounced linewidth narrowing of vertical metallic split-ring resonators via strong coupling with metal surface” (journal name: Nanomaterials)
The authors theoretically demonstrate plasmonic coupling between magnetic plasmon resonance and surface plasmon polaritons, resulting in ultranarrow band hybrid mode and significantly enhanced magnetic field. The proposed metamaterial has high sensitivity and figure-of-merit. The results are interesting and the manuscript is well written. However, the following issues should be resolved before recommending for publication.
- Index sensing by using metasurfaces and nanostructures has been extensively investigated over the past decade. As the major parameters representing the performance as index sensor, in the main text the values of index sensitivity S and figure-of-merit (FoM) should be compared with preceding results. In this way, the authors have to demonstrate the advantage of the proposed vertical split-ring resonators supported by metallic substrates.
- There are many preceding results reported significant increase of field enhancement and localization by using metallic substrates. Metallic substrates-assisted field enhancement based on different mechanisms have been reported in the literature: Light Sci. Appl. 2018, 7, 44; Phys. Chem. Chem. Phys. 2019, 21, 19076-19082. The authors should cite the above references and comparatively describe the differences from the above results.
Reviewer 3 Report
The article entitled "Pronounced linewidth narrowing of vertical metallic split-ring resonators via strong coupling with metal surface" is devoted to a theoretical study of a 3D material consisting of vertical Au split-ring resonators array on Au substrate. Since such devices have great potential for practical application, this article is interesting and relevant. The main conclusions of the article were confirmed by theoretical calculations, and the obtained results are promising. My comments are given below:
1) The introduction can be improved. The articles related to potential applications of split-ring resonators in biosensing should be added.
2) It would be better to add references to the results of previous works related to the study of split-ring resonators (for example, on a silica substrate) in the section devoted to investigating the performance of the Au VSRRs on Au substrate for optical sensing. It will allow comparing the results of previous studies with those obtained in this work.
